# Spin-valley coupling in single-electron bilayer graphene quantum dots

L. Banszerus [1,2✉], S. Möller [1,2], C. Steiner[1,2], E. Icking[1,2], S. Trellenkamp[3], F. Lentz[3], K. Watanabe [4], T. Taniguchi [5], C. Volk [1,2] & C. Stampfer [1,2]

Understanding how the electron spin is coupled to orbital degrees of freedom, such as a valley degree of freedom in solid-state systems, is central to applications in spin-based electronics and quantum computation. Recent developments in the preparation of electrostatically-confined quantum dots in gapped bilayer graphene (BLG) enable to study the low-energy single-electron spectra in BLG quantum dots, which is crucial for potential spin and spin-valley qubit operations. Here, we present the observation of the spin-valley coupling in bilayer graphene quantum dots in the single-electron regime. By making use of highly-tunable double quantum dot devices we achieve an energy resolution allowing us to resolve the lifting of the fourfold spin and valley degeneracy by a Kane-Mele type spin-orbit coupling of $\approx 60\,\mu$eV. Furthermore, we find an upper limit of a potentially disorder-induced mixing of the $K$ and $K'$ states below $20\,\mu$eV.

[1] JARA-FIT and 2nd Institute of Physics, RWTH Aachen University, Aachen, Germany. [2] Peter Grünberg Institute (PGI-9), Forschungszentrum Jülich, Jülich, Germany. [3] Helmholtz Nano Facility, Forschungszentrum Jülich, Jülich, Germany. [4] Research Center for Functional Materials, National Institute for Materials Science, Tsukuba, Japan. [5] International Center for Materials Nanoarchitectonics, National Institute for Materials Science, Tsukuba, Japan. ✉email: luca.banszerus@rwth-aachen.de

The valley pseudospin is an inherent property of two-dimensional honeycomb crystals and - together with the electron spin - makes graphene and bilayer graphene (BLG) interesting for applications in spin- and valley-based electronics and quantum computation[1,2]. This pseudospin arises from the orbital degree of freedom of the independent energy valleys located at the inequivalent vertices ($K$ and $K'$) of the hexagonal Brillouin zone[3]. In analogy to the real spin, the valley pseudospin exhibits also a valley Zeeman effect[4–6], where the valley Zeeman splitting – varying linearly with (out-of-plane) magnetic field – is a result of the orbital magnetic moments originating from the non-vanishing Berry curvature, $\Omega$, at the K-points of gapped BLG (see Fig. 1a). Since these magnetic moments, which have opposite signs for the two valleys, crucially depend on the wave function, the valley $g$-factor in BLG quantum dots can be tuned by electric fields[7,8], offering promising and interesting possibilities for manipulation. However, to fully exploit the potential to manipulate and control both the valley and spin degrees of freedom in BLG quantum dots (QDs), a detailed understanding of their interaction is essential. This is as relevant for a better understanding of spin decoherence processes as it is for exploring ways to electrically manipulate the spin degree of freedom via spin-orbit interaction and implementing innovative spin-valley qubits[2]. Indeed, a detailed understanding of the low-energy spectrum of single particle states within the first electronic orbital (see Fig. 1b) is crucial for finding suitable working points and manipulation mechanisms for possible qubit operation.

Although the single-particle spectrum in BLG QDs has been intensively studied in recent years[9–11], the low-energy spin-valley coupling in BLG QDs has remained experimentally unexplored. This is certainly partly due to the high energy resolution required, as theoretical studies predict an intrinsic spin-orbit (SO) coupling in graphene and BLG of around $\Delta_{SO} \approx 24\,\mu eV$[12–16] and only recently, experiments have – partly indirectly – reported values in the range between 40 and 80 $\mu eV$[17,18]. Moreover, our current knowledge with respect to a possible mixing of $K$ and $K'$ states is very limited. The latter is expressed by $\Delta_{KK'}$ and could allow to access helical states[19].

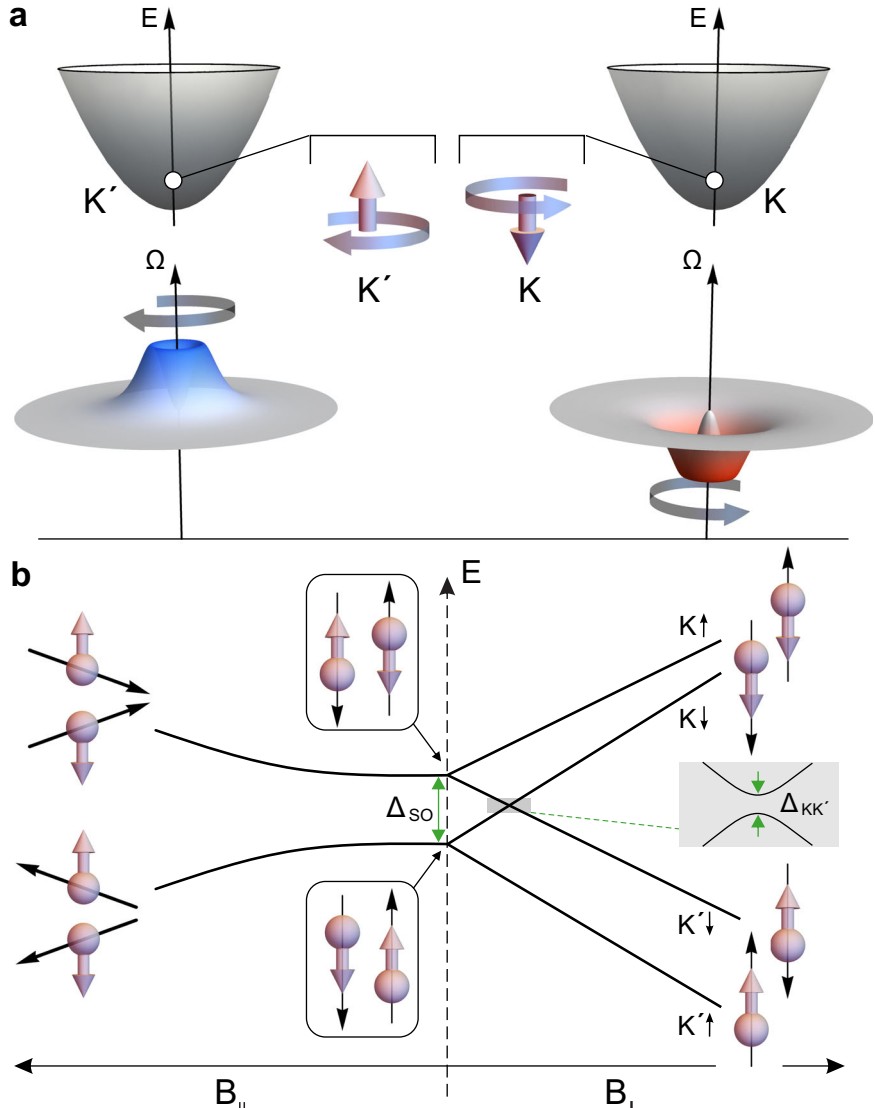

**Fig. 1 Band structure and single particle spectrum of a BLG quantum dot. a** Low energy band schematic of BLG at the $K$ and $K'$ points. BLG exhibits a non-trivial Berry curvature $\Omega$ that leads to an effective out-of-plane magnetic moment with opposite sign at $K$ and $K'$. **b** Energy dispersion of single-particle states in BLG QDs as a function of in-plane ($B_{\parallel}$, left) and out-of-plane ($B_{\perp}$, right) applied magnetic fields with respect to the BLG plane. The SO gap, $\Delta_{SO}$, lifts the fourfold degeneracy and polarizes the spins out-of-plane for zero magnetic field and a potential $K$-$K'$ state mixing (described by $\Delta_{KK'}$) leads to an anticrossing of the $K\downarrow$ and $K'\downarrow$ state.

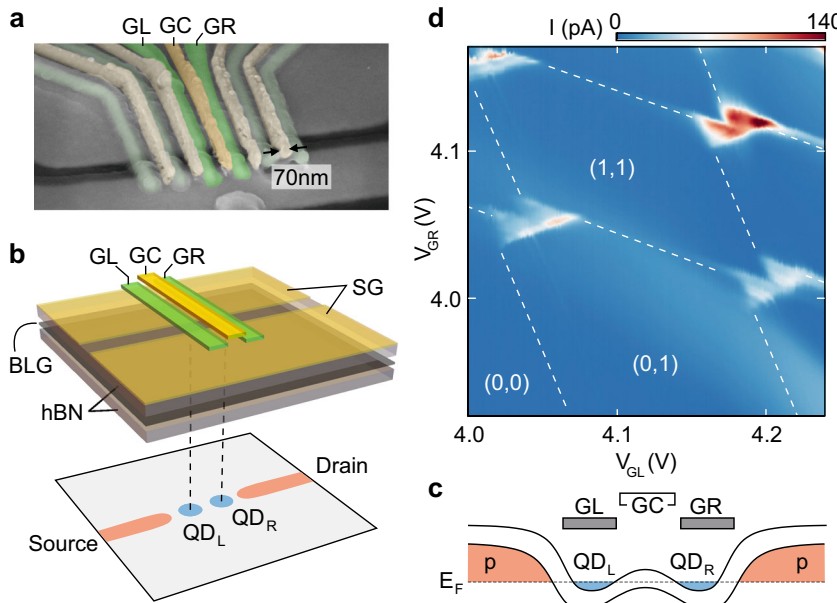

**Fig. 2 Device tuning. a** False-color scanning electron micrograph of the metallic gates. A pair of split gates defines the conducting channel, which can be modulated by voltages applied to the finger gates. The gates used in the following are color coded. **b** Schematic cross-section of the device. The upper part shows the metallic gates on top of the hBN/BLG/hBN van-der-Waals heterostructure. The lower part color codes the charge carrier density within the channel and the two quantum dots (red: holes and blue: electrons). **c** Schematics of the band edge profile along the narrow channel, highlighting how the finger gates are used to form a DQD consisting of $QD_L$ and $QD_R$ connected to the p-type conducting channel. **d** Charge stability diagram showing the current through a double quantum dot as function of the potential applied to the two gate fingers, $V_{GL}$ and $V_{GR}$. A constant bias voltage of $V_b = 1$ mV is applied and the central finger gate voltage is kept at $V_{GC} = -4$ V.

In this letter, we report on measurements of the excited state spectrum of single-electron double quantum dots (DQDs) in BLG providing information on $\Delta_{SO}$ as well as on $\Delta_{KK'}$. By tuning a DQD to a regime of low interdot tunnel coupling, we are able to resolve the interdot transitions with remarkably high energy resolution allowing to reconstruct the underlying single particle spectrum of both quantum dots. We find that the spin and valley degeneracy of the single particle spectrum is lifted by a Kane–Mele type SO gap[13] of $\Delta_{SO} \approx 60\,\mu\text{eV}$, which separates the two Kramer's pairs – $(K' \uparrow, K \downarrow)$ and $(K' \downarrow, K \uparrow)$ – similar (but smaller in magnitude) to what has been observed in carbon nanotube QDs[20,21]. The disorder-induced mixing of $K$ and $K'$ states is found to be at least smaller than $\Delta_{KK'} < 20\,\mu\text{eV}$, where the upper bound is resulting from the energy resolution of our measurements. Figure 1b depicts the first four BLG QD states composing the first electronic orbital ("shell") as a function of the magnetic fields applied in-plane ($B_{\parallel}$) and out-of-plane ($B_{\perp}$) to the BLG sheet. At zero magnetic field, the four states are split into two Kramer's pairs, separated by $\Delta_{SO}$. Applying an out-of-plane magnetic field linearly shifts the energy of the states according to the spin and valley Zeeman effects $E(B_{\perp}) = \frac{1}{2}(\pm g_s \pm g_v)\mu_B B_{\perp}$, where $\mu_B$ is the Bohr magneton and $g_v$ is the valley g-factor, which quantifies the strength of the valley magnetic moment. Note, that $g_v$ strongly depends on the QD's wave function and thus on the size of the QD[11]. It is usually one order of magnitude larger than the spin g-factor, $g_s = 2$. As the valley magnetic moment is oriented perpendicular to the BLG plane, in-plane B-fields only couple to the electron spin. However, as the SO coupling acts as an effective out-of-plane magnetic field close to the K-points, the spin states are polarized perpendicular to the BLG plane (see insets in Fig. 1b)[12]. Applying an in-plane magnetic field therefore shifts the states according to $E(B_{\parallel}) = \pm\frac{1}{2}\sqrt{\Delta_{SO}^2 + (g_s\mu_B B_{\parallel})^2}$, recovering the linear spin Zeeman effect for high B-fields.

## Results

**Device characterization.** The devices consist of a BLG flake, which has been encapsulated between two ($\approx 25$ nm thick) flakes of hexagonal boron nitride (hBN) and has been placed on a graphite flake, acting as a back gate (BG), using a dry van-der-Waals stacking technique. Cr/Au split gates (SGs) are deposited on top, forming a $2\,\mu\text{m}$ long and 130 nm wide channel. Two layers of metallic Cr/Au finger gates (FGs) with a width of 70 nm and a pitch of 150 nm are fabricated across the channel. Details on the fabrication process can be found in ref. [6]. Figure 2a shows a scanning electron micrograph of the gate structure, where the gates used as plunger gates are color coded. Figure 2b shows a schematic cross section through the heterostructure and the gate stack highlighting the formation of the QDs and source-drain regions by electrostatic soft-confinement. All measurements are performed in a helium dilution refrigerator at a base temperature of 10 mK, using standard DC measurement techniques.

QDs are created using three layers of top gates, following previous studies of gate-defined BLG QDs[5,6,9,10,22,23]. A band gap is opened by applying an out-of-plane displacement field[24,25] with the help of the SG ($V_{SG} = 1.73$ V) and BG ($V_{BG} = -1.56$ V), while the Fermi energy ($E_F$) is tuned into the band gap. This leaves a narrow p-type conducting channel, connecting source and drain. A single electron DQD can be formed using adjacent FGs on the lower FG layer (GL and GR), locally overcompensating the BG voltage (see lower illustration in Fig. 2b and the band edge diagram in Fig. 2c, highlighting the potential landscape along the narrow channel)[6]. By applying $V_{GC} = -4$ V to the central FG between GL and GR, the interdot tunnel coupling is reduced in order to enhance the energy resolution of the bias spectroscopy measurements. Figure 2d shows a charge stability diagram of the first four pairs of triple points (see Supplementary Fig. 1 for more details).

Next, we focus on the (0,1)–(1,0) charge transition, where each of the QDs is at most occupied by a single electron. Importantly,

the combined tunneling rate is reduced to $\Gamma < 1$ GHz by GC on the upmost gate layer (see Fig. 2a–c). This reduces the tunnel broadening of the resonance lines and strongly suppresses transport if the states in the two QDs are off resonance.

**Magneto-transport spectroscopy.** Figure 3a–c show finite bias charge stability diagrams of the first triple point pair at out-of-plane magnetic fields of $B_\perp = 0$, 0.2 and 0.4 T. In order to describe the configurations of the DQD, we introduce the orthogonal axes $\delta$ and $\varepsilon$, which describe how far the states in both QDs are tuned into the bias window ($\delta$) and how large their energy detuning ($\varepsilon$) is, respectively. Note, that in the single electron regime, the QD transitions (chemical potentials) are equivalent to the single particle energies. At zero $B$-field, two resonances close to zero detuning (resonances (i) and (ii), (iii)) are visible, while the rest of the triple point shows only suppressed current. Increasing $B_\perp$ shifts one of the resonances (ii) to higher detuning (compare green arrows in Fig. 3a–c). Eventually, a third resonance appears (iv), which does not extend as far on the $\delta$-axis as the other transitions (see Fig. 3c).

The nature of these resonances can be explained in terms of transitions from single-particle states in the left QD (QD$_L$) to single-particle states in the right QD (QD$_R$). For the present interdot tunneling times ($\approx 10$ ns), we assume that the electron spin is entirely conserved, while phonon-assisted valley relaxation may occur on these time scales, as well as during interdot tunneling[26]. We consider the combined tunneling rate, $\Gamma_{comb}$ to be limited by the interdot tunneling rate, $\Gamma_m$: $I/e = \Gamma_{comb} \approx \Gamma_m$, where $I$ is the current through the DQD device and $e$ the elementary charge. This is supported by the absence of any $\delta$ dependence of the transitions. Figure 3d shows a line cut through the triple point in Fig. 3a along the yellow dashed line. The two transitions (black arrow and green arrows in Fig. 3a) result in two distinct peaks in the tunneling current. The first resonance, (i) occurs at $\varepsilon = 0$, where every state in the left QD can tunnel into its equivalent state in the right QD, highlighted by the black arrow (see left schematic in Fig. 3d). The second resonance occurs at $\varepsilon = \Delta_{SO} = 68 \pm 7\ \mu eV$, where two processes are possible, both requiring valley flips, namely transition (ii): $(K' \uparrow)_L \Rightarrow (K \uparrow)_R$ and transition (iii): $(K \downarrow)_L \Rightarrow (K' \downarrow)_R$, highlighted by the two green arrows (see right schematic in Fig. 3d).

When applying an out-of-plane $B$-field, the energies of the single particle states shift according to their spin and valley Zeeman effect, as depicted on the right-hand side of Fig. 1b. Consequently, the detuning energy necessary for the transition (ii) increases linearly with magnetic field, highlighted by the light green arrow in Fig. 3e. The observed increase in detuning energy corresponds to a valley $g$-factor of $g_\nu \approx 15$. This observation validates the assumption that valley flips are allowed, since otherwise interdot transitions should not shift as function of $B_\perp$. The detuning required for transition (iii), decreases to zero, once the involved states are equal in energy, which is the case at $B_\perp \approx 0.18$ T (see Fig. 3e). At higher magnetic fields, the reversed process, transition (iv): $(K' \downarrow)_L \Rightarrow (K \downarrow)_R$ becomes possible, which also shifts with a valley $g$-factor of $g_\nu \approx 15$. This transition is marked by the orange arrow in Fig. 3c and offset to the transition (ii) by $\Delta\varepsilon = 2\Delta_{SO}$, which becomes apparent from Fig. 3e. Since transition (iv) originates from an excited state (ES) in the QD$_L$, it only becomes accessible as soon as the ES enters the bias window. Therefore, the transition line (iv) has a shorter extent along the $\delta$-axis, as it only sets in at finite $\delta$, which is highlighted by the black line in Fig. 3c. The observation of transition (iv) also justifies the assumption of spin conservation. If the spin lifetime would be shorter than the tunneling rate, $(K' \downarrow)_L$ would decay into $(K' \uparrow)_L$,

blocking this process and the resonance would not be visible. Please note that the tunneling current corresponding to transitions (ii)-(iv) is energy ($\varepsilon$) and $\delta$-dependent as resonant tunneling (in particular with the drain reservoir) leads to higher tunneling currents (see Fig. 3a–c).

Figure 4 shows the interdot transitions as function of in-plane (Fig. 4a, b) and out-of-plane (Fig. 4c, d) $B$-field highlighting the spin and valley texture of the low energy spectrum. In Fig. 4a, we show the derivative of the tunneling current $I$ with respect to $\varepsilon$ as function of $\varepsilon$ and $B_\parallel$. Apart from the zero detuning transition (horizontal dashed line), one additional feature is visible (curved dashed line), which corresponds to the transitions from the energetically lower Kramer's pair to the energetically higher Kramer's pair, namely $(K' \uparrow)_L \Rightarrow (K \uparrow)_R$ and $(K \downarrow)_L \Rightarrow (K' \downarrow)_R$, as highlighted by the schematic insets of Fig. 3d. Increasing the in-plane $B$-field increases the energy difference between the Kramer's pairs due to the spin Zeeman effect and therefore the required detuning shifts according to $\varepsilon = \sqrt{\Delta_{SO}^2 + (g_s \mu_B B)^2}$.

Fitting this equation with fixed $g_s = 2$, which is in good agreement with earlier measurements[22], incl. electron spin resonance experiments[27], yields $\Delta_{SO} = 62 \pm 6\ \mu eV$. This results in the dashed and solid lines in Fig. 4a, b showing good agreement with the experiment. Applying an in-plane magnetic field tilts the spin from an out-of-plane orientation induced by the SO coupling into the plane of the BLG (see insets in Fig. 4b). This effect continuously reduces the overlap between the spin states from different Kramer's pairs, e.g., the spin state from $K' \uparrow$ is not perfectly parallel to $K \uparrow$ anymore, until for large $B_\parallel$ they will eventually be completely orthogonal. This effect becomes visible in the tunneling current of the ES transition, which decreases with increasing in-plane magnetic field. In the top panel of Fig. 4b, we show the ratio between the current through the exited and the ground state ($\varepsilon = 0$) as a function of $B_\parallel$ highlighting this effect, which is in good agreement with what is expected from theory (see dashed line and figure caption).

Figure 4c shows the derivative of the tunneling current with respect to $\varepsilon$, as function of energy and $B_\perp$. Here, according to Fig. 1b the transition spectrum is significantly richer and all three transitions (ii), (iii) and (iv) discussed in Fig. 3e can be observed (see dashed lines and labels in Fig. 4d). If we denote the valley $g$-factor in the left and right QD with $g_{\nu,L}$ and $g_{\nu,R}$, respectively, then we can express the different transition energies as a function of $B_\perp$ by $\varepsilon_{ii} = \Delta_{SO} + \frac{1}{2}(g_{\nu,R} + g_{\nu,L})\mu_B B_\perp$, $\varepsilon_{iii} = \Delta_{SO} - \frac{1}{2}(g_{\nu,R} + g_{\nu,L})\mu_B B_\perp$, $\varepsilon_{iv} = -\Delta_{SO} + \frac{1}{2}(g_{\nu,R} + g_{\nu,L})\mu_B B_\perp$. Here, the electron spin Zeeman effect has no influence on the transition energies, as transitions only occur between states of the same electron spin. If that would not be the case, many more transitions would be possible, e.g., $(K' \downarrow)_L \Rightarrow (K' \uparrow)_R$, which would yield a line originating at $\varepsilon = \Delta_{SO}$ with much flatter slope of $g_s = 2$. Taking $\Delta_{SO}$ from the analysis of the in-plane $B$-field data (Fig. 4a, b) and choosing the valley $g$-factors to be $g_{\nu,L} = g_{\nu,R} = 15$ we find – without any additional parameter – good agreement with the experimental data (see dashed and solid lines in Fig. 4c, d). The values of $g_\nu$ are similar to those reported in previous studies of similar BLG QDs and compatible with theoretical calculations[6,23]. Considering the same geometry of both QDs and the similar voltages applied to GL and GR, it is reasonable to assume that both QDs have very similar valley $g$-factors. Note, that as soon as a transition requires negative detuning, it becomes Coulomb blockaded, which is the reason why no transition is observed below $\varepsilon = 0$. The lack of clear signatures of the transition (ii) at low $B$-field (see Fig. 4c, d) can be explained by their reduced tunneling current compared to transition (iii) due to the stronger detuning of the $(K' \uparrow)_L$ ground state from the source chemical potential compared to $(K \downarrow)_L$, combined with strong resonant tunneling from source to the left QD.

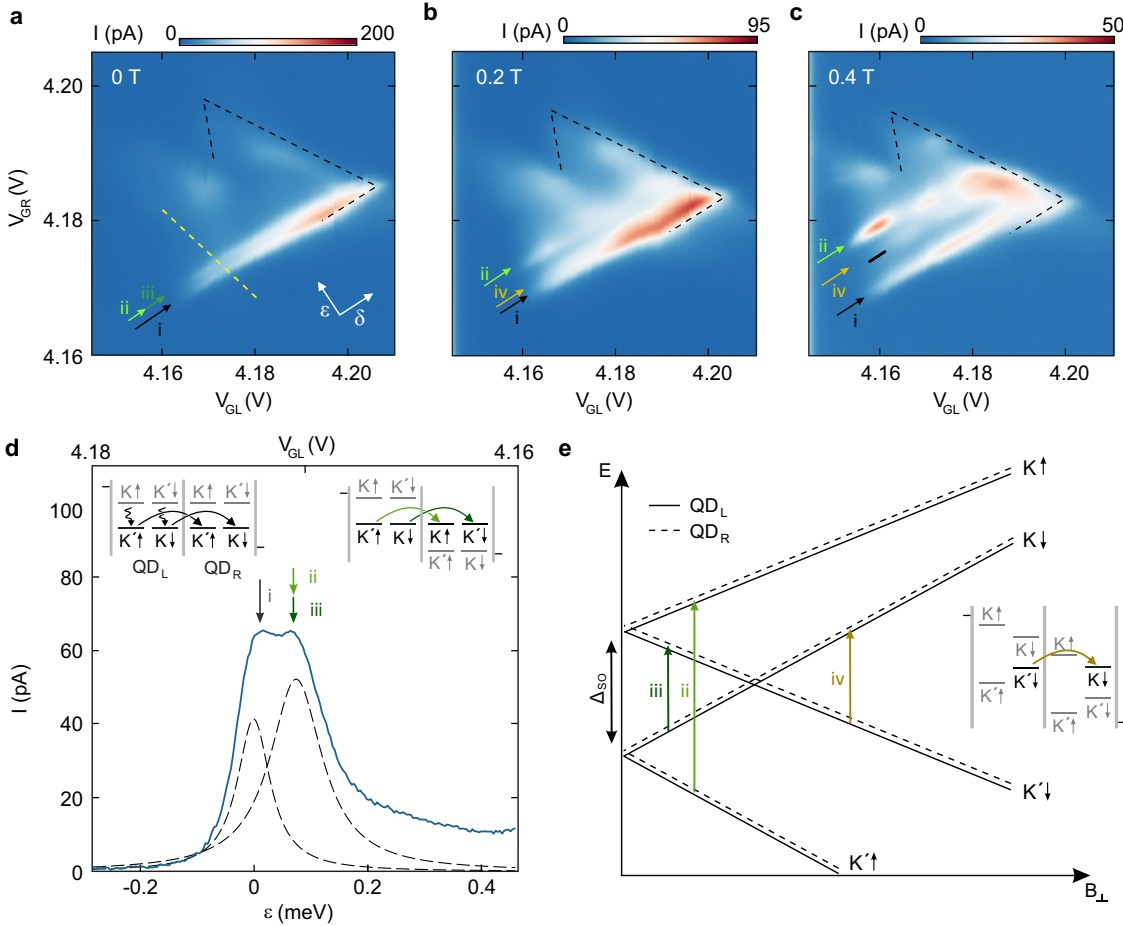

**Fig. 3 (0,1)-(1,0) triple point.** Charge stability diagrams of the first pair of triple points ((1,0)-(0,1) transition) measured at $V_b = 1$ mV and **a** $B_\perp = 0$ T, **b** $B_\perp = 0.2$ T, **c** $B_\perp = 0.4$ T. The white arrows in panel **a** indicate the orthogonal detuning axis $\varepsilon$ and the DQD's common energy $\delta$. At zero magnetic field, two transition lines are visible (green and black arrows). **d** Cut along the yellow dashed line in the first triple point at $B = 0$ T. Two Lorentzian peaks are fit to the data to extract a splitting of $\Delta_{SO} = 68 \pm 7\,\mu$eV. Inset: Schematic energy diagrams of a DQD in the finite bias regime for different interdot detuning energies $\varepsilon$, illustrating resonant transport through the ground state of each QD (transition (i); left inset) and resonant transport at $\varepsilon = \Delta_{SO}$ (transitions (ii) and (iii); right inset). **e** Schematic of the single particle energy spectra of the first orbital of each QD as a function of $B_\perp$. The arrows (color coding as in panels **a–c**) indicate spin conserving transitions from single particle states in the left QD (solid lines) to single particle states in the right QD (dashed lines). The inset shows the transition (iv) at finite magnetic field.

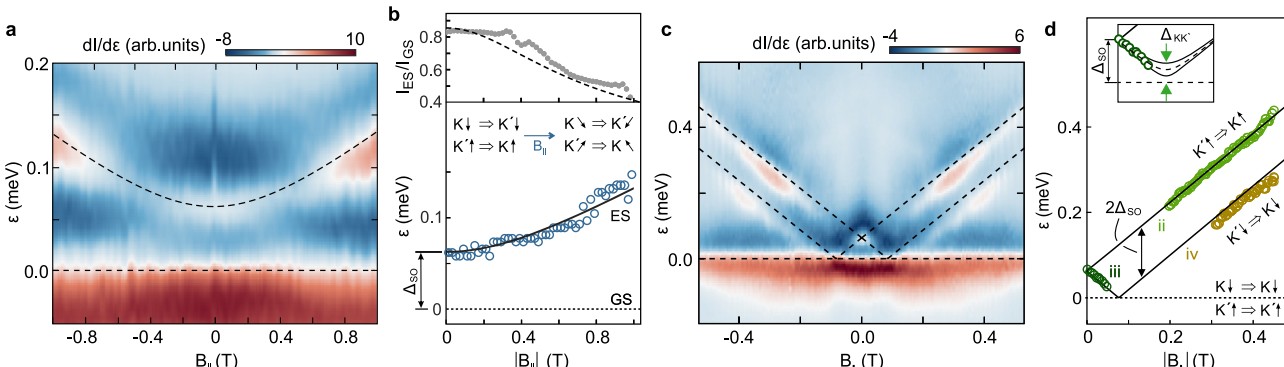

**Fig. 4 Magnetospectroscopy measurements. a** Magnetotransport measurements showing $dI/d\varepsilon$ as function of detuning energy and in-plane $B$-field. The horizontal dashed line marks the ground state (GS) transition ($\varepsilon = 0$) and the curved line marks the excited state (ES) transition. **b** The upper panel shows the ratio of the ES current with respect to the GS current. From a simple projection model, we obtain the dashed line, which describes the expected decay in amplitude of the ES due to the spins tilting in-plane. **c** Similar data as in panel **a** but for out-of-plane $B$-field. The dashed lines mark all transitions described in Fig. 3e. **d** Extracted transition energies as function of $B_\perp$ highlighting the transitions (ii), (iii) and (iv) (see labels). The inset shows a low-energy close-up around the transition (iii) highlighting the presence of a finite $K - K'$ mixing. The dashed line corresponds to transition energies, including a $\Delta_{KK'} = 20\,\mu$eV. The solid lines below and above correspond to $\Delta_{KK'} = 10\,\mu$eV and $30\,\mu$eV, respectively.

By closely inspecting the transition (iii), especially close to the $B$-field regime where the $K \downarrow$ state is crossing the $K' \downarrow$ state (i.e., around $\varepsilon = 0$), we can provide an estimate of the upper limit of a possible disorder-induced mixing of the $K$ and $K'$ states. The inset of Fig. 4d shows a close up, where we included the expected transition energies for different values of $\Delta_{KK'}$ (see also Supplementary Fig. 3). We do not observe any anticrossing within the margin of the energy resolution of our measurement, neither for the device presented in Fig. 4d nor for a second single-electron DQD device presented in Supplementary Fig. 2. From this comparison, we estimate that $\Delta_{KK'}$ is surely not exceeding a value of $20\,\mu eV$ in both DQD devices. Note, that this upper limit is significantly smaller than for carbon nanotubes with values on the order of $100\,\mu eV$[20,21]. In carbon nanotubes the present $\Delta_{KK'}$ may also influence the magnitude of the zero-field splitting of the Kramer's pairs and thus potentially lead to an overestimation of $\Delta_{SO}$. As the zero field splitting consists of the quadratic sum of the two effects, $\sqrt{\Delta_{SO}^2 + \Delta_{KK'}^2}$, and our observed splitting is at least a factor of three larger than $\Delta_{KK'}$, the influence of the intervalley mixing on the observed values of $\Delta_{SO}$ is smaller than 10% and thus lies within the range of our measurement uncertainties.

## Discussion

Apart from the Kane–Mele SO coupling, which is intrinsically present in graphene and BLG, extrinsic Bychkov–Rashba SO coupling and pseudospin inversion asymmetry (or principal plane asymmetry) SO coupling can in principle also play a role[12–16,28,29]. The latter, which arises for example when placing graphene or BLG on substrates, such as e.g., hBN, depends on the magnitude of $k$ (measured from the corners of the Brillouin zone) and is thus suppressed at the $K$ and $K'$-points[16], where our devices are operated. The Rashba-type SO coupling needs to be discussed in more detail, since in our devices the inversion symmetry is explicitly broken by the applied out-of-plane displacement field. The breaking of inversion symmetry and the magnitude of the resulting Bychkov–Rashba SO gaps have theoretically been investigated by Kane and Mele for single-layer graphene[13] and by Konschuh et al. for BLG[12]. Both studies conclude that the Rashba-type SO coupling is negligible ($\approx 1\,\mu eV$) compared to the Kane–Mele coupling term. In addition, the Bychkov–Rashba SO coupling term is expected to be strongly suppressed in BLG single-electron QDs, since specifically in BLG this term vanishes for the low energy bands close to the $K$ and $K'$-points[12]. Thus, all this is expected to lead to a displacement-field-independent SO gap that can be experimentally verified.

In Fig. 5, we show the SO gap as function of displacement field, $D$. Here, $\Delta_{SO}$ has been extracted from zero $B$-field data similar to the measurements shown in Fig. 3a, d but with different back and split gate voltages such that the $D$-field is tuned from $D = 0.24\,V/nm$ to $0.34\,V/nm$, also resulting in different band gaps in the BLG as highlighted by the insets in Fig. 5. From all data presented in Fig. 5 – including data from a second single-electron DQD device (red triangles) and data from a different single-electron QD device (yellow square, more details Supplementary Fig. 4) – we conclude that the observed SO gaps are all consistent and within the error bars constant over the investigated $D$-field range, with a mean value around $\Delta_{SO} \approx 60\,\mu eV$. From the absence of any dependency of $\Delta_{SO}$ as function of the strength of the potential breaking the inversion symmetry, we conclude that the experimentally extracted $\Delta_{SO}$ is dominated by the Kane–Mele coupling term. Interestingly, our $\Delta_{SO}$ values are slightly larger than what has been extracted in previous experiments performed in bulk graphene on trenched $SiO_2$[18]. Also this value is larger than theoretically predicted[12], but might be explained by an enhancement due to phonon-assisted SO coupling[30]. In our case we expect that

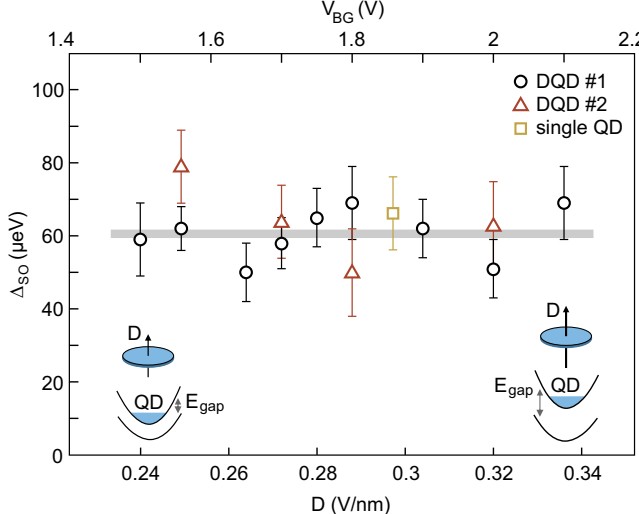

**Fig. 5 Spin-orbit gap $\Delta_{SO}$ as function of applied electric displacement field, $D$.** The black data points correspond to the device presented in the main manuscript (DQD #1) and the red data points to the second device (DQD #2) shown in Supplementary Fig. 2. The yellow data point is obtained from a single quantum dot (single QD) formed in a third device (see Supplementary Fig. 4). The errorbars represent the measurement uncertainties resulting from finite linewidths in the measurements. Over the entire displacement field range, $\Delta_{SO} \approx 60\,\mu eV$ remains constant within the margin of uncertainty (see gray horizontal line). Note that the $V_{BG}$-scale (upper horizontal axis) is valid only for the two double QD devices but slightly off for the single QD. Inset: Schematic illustration of the band structure of BLG with a small and a large applied out-of-plane displacement field, which breaks the inversion symmetry and leads to the opening of a band gap ($E_{gap}$) while $\Delta_{SO}$ remains unaffected.

the SO coupling is slightly enhanced due to the proximity effect when encapsulating BLG with hBN crystals[17], very similar to the proximity enhanced SO coupling when placing BLG on $WSe_2$[31].

In summary, we studied the low-energy excited state spectrum of a gate-defined single-electron quantum dot in bilayer graphene. We find a spin-valley coupling dominated by a Kane–Mele type SO coupling with $\Delta_{SO} \approx 60\,\mu eV$, giving rise to two Kramer's pairs with either parallel or anti-parallel spin-valley orientation. The small value for $\Delta_{KK'}$ ($< 20\,\mu eV$) is not entirely unexpected for flat and disorder-free BLG, and raises the hope that the existing spin-valley coupling – without $K - K'$ mixing – will be helpful for a future qubit operation.

## Data availability

The data supporting the findings are available in a Zenodo repository under https://doi.org/10.5281/zenodo.5258323.

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

## Acknowledgements
The authors acknowledge discussions with J. Fabian, A. Knothe and V. Fal'ko and thank J. Klos for help with the SEM micrographs. This project has received funding from the European Union's Horizon 2020 research and innovation programme under grant agreement No. 881603 (Graphene Flagship) and from the European Research Council (ERC) under grant agreement No. 820254, the Deutsche Forschungsgemeinschaft (DFG, German Research Foundation) under Germany's Excellence Strategy - Cluster of Excellence Matter and Light for Quantum Computing (ML4Q) EXC 2004/1 - 390534769, through DFG (STA 1146/11-1), and by the Helmholtz Nano Facility[32]. Growth of hexagonal boron nitride crystals was supported by the Elemental Strategy Initiative conducted by the MEXT, Japan, Grant Number JPMXP0112101001, JSPS KAKENHI Grant Numbers JP20H00354 and the CREST(JPMJCR15F3), JST.

## Author contributions
L.B., S.M., E.I., S.T. and F.L. fabricated the device, L.B., S.M., C.Ste. and C.V. performed the measurements and analyzed the data. K.W. and T.T. synthesized the hBN crystals. C.V. and C.Sta. supervised the project. L.B., S.M., C.Ste., C.V. and C.Sta. wrote the manuscript with contributions from all authors. L.B. and S.M contributed equally to this work.

## Funding

## Competing interests
The authors declare no competing interests.
