## [Peer Review File · Nature Communications]

REVIEWER COMMENTS

Reviewer #1 (Remarks to the Author):

Spin-valley coupling in single-electron bilayer graphene quantum dots

This manuscript explores double quantum dots in bilayer graphene and reports lifting of the fourfold spin and valley degeneracy in the first electronic orbital, which sets the stage for controlled manipulation of spin and valley degrees of freedom. By investigating the excited state spectrum of double quantum dots with high energy resolution, the authors extract energy scales for the Kane-Mele spin-orbit gap and the valley Zeeman splitting with magnetic field. The double quantum dot devices that are reported in this paper are some of the cleanest and most technically sophisticated that have been realized in bilayer graphene to date, which represents a significant step towards realization of graphene-based device platforms for quantum information processing.

Valley Zeeman splitting with magnetic field has already been reported in the literature in graphene quantum dots, so the primary novel claim of this paper is the observation of a lifting of the fourfold spin and valley degeneracy via spin-orbit coupling. However, in previous experiments on carbon nanotube quantum dots, disorder-induced mixing of the K and K' states have obscured the effects of spin-orbit coupling. In this manuscript, the energy scale of the disorder-induced K, K' degeneracy lifting is close in magnitude to the spin-orbit gap measured in microwave experiments (J. Sichau et al. PRL 122, 046403 (2019)), which somewhat weakens the author's claims that spin-orbit effects are unambiguously observed in their double quantum dot system.

To demonstrate the intrinsic nature of the reported phenomena, this paper would be significantly strengthened by: (i) showing reproducibility of these results in additional double quantum dot devices and (ii) demonstrating that the spin-orbit splitting can be resolved in the excitation spectra of each individual dot.

Questions for the authors:

(1) Could you provide additional data sets to demonstrate reproducibility of the excitation spectra results in additional quantum dot devices, in order to establish the intrinsic nature of the reported phenomena? How do the results depend on the shape of the electronic confinement potential?

(2) As a consistency check, have you performed excited state spectroscopy of a single electron in each dot separately?

(3) Small comment: The cross-sectional device schematic and band structure plot in Fig 1a,b of the Supplementary Information is helpful for understanding the device structure. Would it be possible to add these panels to Figure 2 (in place of panel b)?

(4) What is the energy resolution of your measurements, and what factors limit this resolution? What is the electron temperature of your device, as extracted from Coulomb blockade fits?

(5) The size of the Kane-Mele spin-orbit gap ($\Delta_{SO} \sim 68$ ueV) reported in this manuscript is significantly smaller than the value of $2\Delta_{SO} \sim 4.2$ K calculated theoretically for graphene (C. L. Kane and E. J Mele, PRL 95, 226801 (2005)) and the value of $\Delta_{SO} = 370$ ueV experimentally reported in carbon nanotubes (F. Kuemmeth et al. Nature 452, 448-452 (2008)). Can you comment on the reason why the gap sizes you observed in BLG differ from these other values reported in the literature?

(6) The paper reports a maximum disorder-induced mixing of the K and K' states of 20 ueV. However, it's unclear whether the Energy vs B plot in Figure 4c has sufficient energy resolution to support this claim. Also, how reproducible is this disorder-induced splitting across multiple samples?

(7) Have you observed a dependence of Δ_{SO} on the occupation of the quantum dots? Also, have you observed any signatures of electron-hole symmetry breaking in the excitation spectra?

Reviewer #2 (Remarks to the Author):

The manuscript contains an experimental study of the low energy spectra in double bilayer graphene quantum dots. The low temperatures (10mK) allow to resolve very low energies. The valley-Zeeman, resulting from the topological orbital magnetic moment, result in large g-factors, which seem promising in the field of valleytronics. The paper is very well written, the graphs clearly illustrate the

effects they measure and the experimental setups are quite well designed, providing good quality results.

However, I do still have a few concerns:

There are other sources of zero-field splitting not included in the model, originated from symmetry breaking potentials, as Rashba or principal-plane asymmetry spin-orbit coupling. However, the experimental setup suggests that these should not be negligible, owing to different symmetry-breaking structures, gates, electric fields, etc.

Could the measured Δ_{SO} correspond to some combination of intrinsic and structural SOC? How does $\Delta_{kk'}$ relates to structural SOC?

It would help to understand why Δ_{SO} differs from the one measured in other experiments.

'Previous experiments' have consistently (since 2012) shown that the intrinsic Δ_{SO} of about 45 micro-eV, which has been reproduced in several other samples using electron-spin resonance.

The inter-valley term is, however, disorder induced or sample dependent, for which an upper bound is given.

That would make this manuscript fall a bit short on novelty, if that was the main focus of the paper.

Minor points:

* ϵ and ϖ are used for the same parameter, it seems.

* orthogonal axes ϵ and δ are introduced but δ is not used in the figures or Ms, rendering it unnecessary.

Comment:

I wonder if the valley-Zeeman effect could have a connection to the relationship of orbital angular momentum and sub-lattice spin in graphene? Or could an intuitive explanation be made, why the valley g-factor is so large?

Reviewer 1:

Spin-valley coupling in single-electron bilayer graphene quantum dots

This manuscript explores double quantum dots in bilayer graphene and reports lifting of the fourfold spin and valley degeneracy in the first electronic orbital, which sets the stage for controlled manipulation of spin and valley degrees of freedom. By investigating the excited state spectrum of double quantum dots with high energy resolution, the authors extract energy scales for the Kane-Mele spin-orbit gap and the valley Zeeman splitting with magnetic field. The double quantum dot devices that are reported in this paper are some of the cleanest and most technically sophisticated that have been realized in bilayer graphene to date, which represents a significant step towards realization of graphene-based device platforms for quantum information processing.

We thank the reviewer for her/his time and effort of reviewing our manuscript and we appreciate the valuable feedback allowing us to improve important aspects of our work. We are pleased to read that the reviewer has a very positive reception of the experiment and considers our work 'a significant step towards realization of graphene-based device platforms for quantum information processing'.

Valley Zeeman splitting with magnetic field has already been reported in the literature in graphene quantum dots, so the primary novel claim of this paper is the observation of a lifting of the fourfold spin and valley degeneracy via spin-orbit coupling. However, in previous experiments on carbon nanotube quantum dots, disorder-induced mixing of the K and K' states have obscured the effects of spin-orbit coupling. In this manuscript, the energy scale of the disorder-induced K, K' degeneracy lifting is close in magnitude to the spin-orbit gap measured in microwave experiments (J. Sichau et al. PRL **122**, 046403 (2019)), which somewhat weakens the author's claims that spin-orbit effects are unambiguously observed in their double quantum dot system.

We thank the reviewer for raising this concern. We agree with the reviewer that the valley Zeeman effect has been reported in literature and that our main claim comprises the detailed understanding of the low energy single particle spectrum in a graphene quantum dot, showing the lifting of the fourfold degeneracy via SO coupling. We kindly disagree with the reviewers concern that the value of the spin orbit (SO) gap, which we report may be dominated or strongly be influenced by $K - K'$ coupling. While the referee is correct in stating that the zero-field splitting of the two Kramer's pairs may be affected if the $K - K'$ coupling is similar in magnitude to the SO gap, the presence of a $K - K'$ coupling also lifts the valley degeneracy of the level spectrum as function of an in-plane magnetic field. This has been observed before in carbon nanotubes, which exhibit a similar spectrum and can be seen e.g. in Fig. 21 and Fig. 22 in *E. Laird et al. Rev. Mod. Phys.* **87** 703 (2015).

A lifting of the valley degeneracy as a function of an in-plane magnetic field is not observed in our experiment, within the energy resolution. Finally, we would like to point out that an upper bound of $\Delta_{KK'} = 20 \mu\text{eV}$ is sufficient to ensure that the zero B-field gap originates almost entirely from the SO gap: The zero-field gap, Δ , in case of finite Δ_{SO} and $\Delta_{KK'}$ is

given by $\Delta = \sqrt{\Delta_{\text{SO}}^2 + \Delta_{\text{KK}'}^2} = 65\mu\text{eV}$. In a 'worst case' scenario, where $\Delta_{\text{KK}'} = 20\mu\text{eV}$ the true SO gap would be given by $\Delta_{\text{SO}} = \sqrt{\Delta^2 - \Delta_{\text{KK}'}^2} = 62\mu\text{eV}$, which lies within the margin of error for the reported value of Δ .

We realize that this important point has not been sufficiently stressed in the manuscript and we have added a paragraph to estimate the influence of $\Delta_{\text{KK}'}$ on the extraction of Δ_{SO} , which now reads (see page 7):

'We do not observe any anticrossing within the margin of the energy resolution of our measurement, neither for the device presented in Fig. 4d nor for a second single-electron DQD device presented in the supplementary information. From this comparison we estimate that $\Delta_{\text{KK}'}$ is surely not exceeding a value of $20\mu\text{eV}$ in both DQD devices. Note that this upper limit is significantly smaller than for carbon nanotubes with values on the order of $100\mu\text{eV}$ [20,21]. In carbon nanotubes the present $\Delta_{\text{KK}'}$ may also influence the magnitude of the zero-field splitting of the Kramer's pairs and thus potentially lead to an overestimation of Δ_{SO} . As the zero field splitting consists of the quadratic sum of the two effects, $\sqrt{\Delta_{\text{SO}}^2 + \Delta_{\text{KK}'^2}$, and our observed splitting is at least a factor three larger than $\Delta_{\text{KK}'}$, the influence of the intervalley mixing on the observed values of Δ_{SO} is smaller than 10% and thus lies within the range of our measurement uncertainties.

To demonstrate the intrinsic nature of the reported phenomena, this paper would be significantly strengthened by: (i) showing reproducibility of these results in additional double quantum dot devices and (ii) demonstrating that the spin-orbit splitting can be resolved in the excitation spectra of each individual dot.

We thank the reviewer for this helpful suggestion and we are happy to adapt our manuscript (i) by adding additional data on the reproducibility of the presented data and (ii) by adding additional, newly recorded single QD excited state spectroscopy data.

Questions for the authors:

(1) Could you provide additional data sets to demonstrate reproducibility of the excitation spectra results in additional quantum dot devices, in order to establish the intrinsic nature of the reported phenomena? How do the results depend on the shape of the electronic confinement potential?

We thank the referee for this suggestion. In Fig. R1 (see below) we present a data set recorded on a second single electron double quantum dot realized under a different set of gate fingers. In Fig. R1a, we present a line cut along the detuning axis of the first triple point at zero magnetic field, similar to Fig. 3d of the main text. As can be seen from the data, two peaks corresponding to transition (i) and transitions (ii) and (iii) can be identified, which are separated by approximately $70\mu\text{eV}$. Fig. R1b and R1c show the same detuning cuts as function of an applied perpendicular magnetic field (c.f. Figure 4c and 4d of the main text). The data in panel b shows the same behaviour as the first device discussed in the main

manuscript. When extracting the transition energies (panel c), a valley g-factor of $g_v = 21$ and a spin orbit gap of $\Delta_{SO} = 79 \mu\text{eV}$ are extracted.

We have added Fig. R1 to the supplementary information accompanying the manuscript in order to demonstrate reproducibility of the results.

A systematic study of Δ_{SO} as function of the shape of the QD is difficult to perform, as the geometry of the QD can only be estimated roughly. We would not expect any dependence of Δ_{SO} on the confinement geometry, as even for larger QDs, the wavefunction is still located very closely around the K-points, where Rashba terms vanish entirely and the SO coupling can be described by the Kane-Mele Hamiltonian alone. Experimentally, we test this by observing the magnitude of Δ_{SO} for a number of different applied electric displacement fields, changing the band gap and thus certainly also the confinement potential of the QD. The data has been included as a new Figure 5 to the main text.

Figure R1: Measurements on a second DQD formed using a different set of gate fingers. **a** Detuning line cut at zero magnetic field (similar measurement as in Fig. 3d of the main text). Two resonances, one corresponding to transition (i) and one corresponding to transitions (ii) and (iii) are observed. The inset shows the triple point, the scale bars correspond to a difference in gate potential of 20 mV. **b** Magnetotransport measurements showing $dI/d\varepsilon$ as function of detuning energy and out-of-plane B -field. The dashed lines mark the observed interdot transitions. **c** Extracted transition energies as function of B_\perp highlighting the transitions (ii), (iii) and (iv) (see labels).

(2) As a consistency check, have you performed excited state spectroscopy of a single electron in each dot separately?

We thank the reviewer for this helpful question. First, we would like to point out that the confinement potential is certainly changed when tuning the device from a DQD to a single QD regime, which may have an influence on the strongly (wavefunction-dependent) valley g-factor, making it hard to directly compare DQD data with single QD data. In Fig. R2a

and R2b, we plot finite bias spectroscopy measurements of the left and right single QD comprising the DQD discussed in the main text, respectively. The data is recorded at a finite perpendicular magnetic field of $B_{\perp} = 0.5$ T, resulting in a sizeable and almost identical valley splitting of $\Delta E \approx 0.8$ meV in both QDs (see dashed lines). Unfortunately, the increased coupling to the source-drain reservoirs leads to a broadening of the single QD states, which does not allow us to observe the (much smaller) spin splitting or even a zero B-field splitting. This is actually the reason why we have chosen – at first place – to perform our study in a DQD, where both thermal broadening, as well as tunnel broadening due to the coupling to the reservoirs does not impact the energy resolution for ‘sharp’ interdot transitions where two discrete states are probing each other. The resulting high energy resolution of interdot transitions have been used before, e.g. in carbon nanotubes (I. Khivrich et al. Nat. Commun. **11** 2299, (2020)).

Figure R2: **a, b** Finite bias spectroscopy measurement of the single electron spectrum of the left (**a**) and right (**b**) single QD of the device discussed in the main text, recorded at an out-of-plane magnetic field of 0.5 T. The valley splitting due to the out-of-plane field can be resolved in the two single QDs and is very similar for both dots. The tunnel broadening of the Coulomb resonances do not allow to resolve the spin splitting. **c** Finite bias spectroscopy measurement performed on a different device on another sample with a design of more opaque tunneling barriers/less tunnel broadening, recorded at a finite perpendicular magnetic field of $B_{\perp} = 0.5$ T. Here, the Zeeman spin splitting can be observed. **d** From similar measurement as depicted in panel c, recorded for various magnetic fields we extract the Zeeman splitting between $|K' \uparrow\rangle$ and $|K' \downarrow\rangle$. From the data, we extract a spin g-factor of $g_s = 2$ and $\Delta_{SO} = 66 \pm 8 \mu\text{eV}$.

To verify that the SO coupling term is present also in single QDs, we present in Fig. R2c a similar finite bias spectroscopy measurement recorded on another BLG QD device (measured in a different setup) which has tunneling barriers with a higher opaqueness. Here, the spin splitting of the $|K' \uparrow\rangle$ and $|K' \downarrow\rangle$ can be observed, whereas $|K \uparrow\rangle$ and $|K \downarrow\rangle$ are almost degenerate as the Zeeman spin splitting cancels out Δ_{SO} . By measuring the energy difference between $|K' \uparrow\rangle$ and $|K' \downarrow\rangle$ for various magnetic fields, we are able to extract a spin g-factor of $g_s = 2$ and a SO gap of $\Delta_{SO} = 66 \pm 8 \mu\text{eV}$ (see Fig. R2d), in very good agreement with the data recorded on the two DQDs.

We agree with the reviewer that providing single particle spectra extracted from finite bias spectroscopy in a single QD is a nice cross-check and we have added Fig. R2 to the sup-

porting information of the manuscript and included the extracted value for Δ_{SO} to the new Figure 5 (see Fig. R5 on page 15 in this document).

(3) Small comment: The cross-sectional device schematic and band structure plot in Fig 1a,b of the Supplementary Information is helpful for understanding the device structure. Would it be possible to add these panels to Figure 2 (in place of panel b)?

We thank the reviewer for this comment. We agree with the referee that especially the band schematic, which was previously displayed in the supplementary information only, would be a helpful addition to Figure 2 of the main text. We modified Fig. 2 such that panel b highlighting the cross section is now enlarged and we added the band alignment schematic as a new panel c to Fig. 2 of the main text.

(4) What is the energy resolution of your measurements, and what factors limit this resolution? What is the electron temperature of your device, as extracted from Coulomb blockade fits?

We thank the reviewer for this question. As briefly mentioned above, the energy resolution of the interdot transitions are hardly affected by the electron temperature or the tunnel coupling to the leads (c.f. I. Khivrich et al. Nat. Commun. **11** 2299, (2020)), as two discrete levels are probing one another. Instead the line broadening results mainly from the interdot tunnel coupling and potentially from charge noise on the bias and gate voltages. From the measurements in Fig. 4 of the main text, we deduce that we are able to resolve energy scales down to approximately $20 \mu\text{eV}$, which also sets the upper bound for $\Delta_{\text{KK}'}$.

While electron temperature cannot be obtained from the (tunnel broadened) finite bias spectroscopy data, we can report an absolute mixing chamber temperature measured with a SQUID thermometer of $T_{\text{MC}} = 11 \pm 1 \text{ mK}$ and we can report an electron temperature of $T_{\text{el}} \approx 100 \text{ mK}$ obtained from the Coulomb peak width of the sample presented in Fig. R2c, measured in an identical setup.

(5) The size of the Kane-Mele spin-orbit gap ($\Delta_{\text{SO}} \approx 68 \text{ ueV}$) reported in this manuscript is significantly smaller than the value of $2\Delta_{\text{SO}} \approx 4.2 \text{ K}$ calculated theoretically for graphene (C.L. Kane and E.J. Mele, PRL 95, 226801 (2005)) and the value of $\Delta_{\text{SO}} = 370 \text{ ueV}$ experimentally reported in carbon nanotubes (F. Kuemmeth et al. Nature 452, 448-452 (2008)). Can you comment on the reason why the gap sizes you observed in BLG differ from these other values reported in the literature?

We thank the referee for raising this question. Before going into details on spin-orbit interaction in graphene and BLG, we would like to clarify that the origin of SO coupling in carbon nanotubes (CNTs) originates mainly from its strong curvature (being absent in graphene), which results in diameter dependent SO gaps on the order of a few hundred μeV up to over one meV in CNTs (see. F. Kuemmeth et al. Nature **452**, 448-452 (2008) and Laird et al. Rev. Mod. Phys. **87** 703 (2015) – in particular section E and F therein). The values of the SO gap

in CNTs can thus not be directly compared to those of flat graphene and BLG.

In contrast, graphene and BLG exhibit only a small intrinsic Kane-Mele spin orbit gap on the order of a few tens μeV . Kane and Mele (C.L. Kane and E.J Mele, PRL **95**, 226801 (2005)) give 'a coarse estimate' of $2\Delta_{\text{SO}} \approx 2.4 \text{ K} \approx 0.2 \text{ meV}$. More accurate band-structure calculations (see S. Konschuh et al. PRB **85**, 115423 (2012)) find an intrinsic Kane-Mele gap of approximately $24 \mu\text{eV}$ for BLG. Both theoretical studies suggest that Rashba-type SO coupling induced by external electric fields should be negligible for meaningful field strengths (1V/nm) for the low energy bands.

So far, experimental studies on single layer graphene on trenched SiO_2 (J. Sichau et al. PRL **122**, 046403 (2019)) have found a value of $\Delta_{\text{SO}} = 42 \mu\text{eV}$ for (non-confined) bulk states. Recent studies on quantum point contacts (one-dimensional confinement) in BLG have found values of $\Delta_{\text{SO}} = 40 - 80 \mu\text{eV}$ (L. Banszerus et al. PRL **124** 1777701 (2020)). An open question is still the discrepancy between the prediction by S. Konschuh et al. and the extracted values for the SO gap in confined BLG nanostructures which all report higher values. Possible explanations could be proximity induced SO coupling via the hexagonal boron nitride crystals on top and below the BLG¹, as well as a phonon mediated increase in the SO gap (H. Ochoa et al. PRB **86** 245411 (2012)) or effects stemming from the confinement potential itself.

We understand now that the comparison to the SO gap of CNTs might be somewhat subtle for the non-expert reader and that explicitly mentioning state-of-the-art theory values is helpful. We therefore adjusted the corresponding paragraph (pg. 6-7) in the manuscript which now reads:

'Apart from the Kane-Mele SO coupling, which is intrinsically present in graphene and BLG, extrinsic Bychkov-Rashba SO coupling and pseudospin inversion asymmetry (or principal plane asymmetry) SO coupling can in principle also play a role [12-16,31,32]. The latter, which arises for example when placing graphene or BLG on substrates, such as e.g. hBN, depends on the magnitude of k (measured from the corners of the Brillouin zone) and is thus suppressed at the K and K' -points [16], where our devices are operated. The Rashba-type SO coupling needs to be discussed in more detail, since in our devices the inversion symmetry is explicitly broken by the applied out-of-plane displacement field. The breaking of inversion symmetry and the magnitude of the resulting Bychkov-Rashba SO gaps have theoretically been investigated by Kane and Mele for single-layer graphene [13] and by Konschuh et al. for BLG [12]. Both studies conclude that the Rashba type SO coupling is negligible ($\approx 1 \mu\text{eV}$) compared to the Kane-Mele coupling term. In addition, the Bychkov-Rashba SO coupling term is expected to be strongly suppressed in BLG single-electron QDs, since specifically in BLG this term vanishes for the low energy bands close to the K and K' -points [12]. Thus, all this is expected to lead to a displacement-field-independent SO gap that can be experimentally verified. In Fig. 5 we show the SO gap as function of displacement field, D . Here, Δ_{SO} has been extracted from zero B -field data similar to the measurements shown in Figs. 3a,d but with different back and split gate voltages such that the D -field is tuned from $D = 0.24 \text{ V/nm}$ to 0.34 V/nm , also resulting in different band gaps in the BLG as

¹Private communication with Jaroslav Fabian.

highlighted by the insets in Fig. 5. From all data presented in Fig. 5 – including data from a second single-electron DQD device (red triangles) and data from a different single-electron QD device (yellow square, more details in the supplementary information) – we conclude that the observed SO gaps are all consistent and within the error bars constant over the investigated D -field range, with a mean value around $\Delta_{\text{SO}} \approx 60 \mu\text{eV}$. From the absence of any dependency of Δ_{SO} as function of the strength of the potential breaking the inversion symmetry, we conclude that the experimentally extracted Δ_{SO} is dominated by the Kane-Mele coupling term. Interestingly, our Δ_{SO} values are slightly larger than what has been extracted in previous experiments performed in bulk graphene on trenched SiO_2 [18]. Also this value is larger than theory predicted [12], but might be explained by an enhancement due to phonon-assisted SO coupling [33]. In our case we expect that the SO coupling is slightly enhanced due to the proximity effect when encapsulating BLG with hBN crystals [17], very similar to the proximity enhanced SO coupling when placing BLG on WSe_2 [34].

(6) The paper reports a maximum disorder-induced mixing of the K and K' states of 20 ueV. However, it's unclear whether the the Energy vs B plot in Figure 4c has sufficient energy resolution to support this claim. Also, how reproducible is this disorder-induced splitting across multiple samples?

We thank the reviewer for raising this concern. The inset of Fig. 4d of the main text compares calculations assuming different values for $\Delta_{\text{KK}'}$ (black lines) to the extracted transition energies (green data points). When comparing the data to the different calculations, we find only for $\Delta_{\text{KK}'} > 20 \mu\text{eV}$ clear deviations from the data, making $\Delta_{\text{KK}'} = 20 \mu\text{eV}$ an upper bound, which is just limited by the measurement resolution. The second DQD device now included in the revised manuscript has a comparable energy resolution allowing to extract also an upper bound of $\Delta_{\text{KK}'} > 20 \mu\text{eV}$.

We agree with the reviewer that the inset in Fig. 4d is rather small. Therefore, we have added a supplementary figure where we present the estimate for $\Delta_{\text{KK}'}$ for both devices (see Fig. R3).

Figure R3: Estimation of the intervalley coupling, $\Delta_{KK'}$, for the two investigated DQD devices (panel **a**: data from device 1 and panel **b**: data from device 2). The blue lines mark the transition energies, where $\Delta_{KK'} = 20 \mu\text{eV}$ is assumed, which marks an upper bound for $\Delta_{KK'}$ in both devices. The figure has been added to the supporting information.

(7) Have you observed a dependence of DeltaSO on the occupation of the quantum dots? Also, have you observed any signatures of electron-hole symmetry breaking in the excitation spectra?

We agree with the reviewer that measuring the occupation of the SO gap at higher QD occupation would be a very interesting experiment. However, extracting the energy scales of the SO gap in a two electron single QD is already non-trivial owing to the complex two-particle spectrum which exhibits 16 possible states, as well as the presence of additional energy scales (long range Coulomb interaction ('exchange'), as well as lattice scale interaction terms, lifting the degeneracy of the orbital symmetric multiplet at zero magnetic field), as described theoretically in details by A. Knothe et al. arXiv:2104.03399). When studying DQDs, the number of possible interdot transitions (even when taking selection rules into account) becomes very large. Therefore, we consider focusing on a two electron DQD somewhat premature at the current level of understanding and would instead focus on fully comprehending the two particle spectrum in a single quantum dot first.

The question of electron-hole symmetry in ambipolar BLG DQDs is in fact a very interesting one. Judging from a theoretical perspective, electron-hole symmetry should entirely be preserved up to intermediate displacement fields, where the BLG hopping elements γ_3 and γ_4 do not interact with the low-energy bands yet and also Rashba type SO coupling (induced by the external electric field) can be entirely neglected. In this situation BLG can be described entirely by the (electron-hole symmetric) 2×2 tight binding Hamiltonian and the (electron-hole symmetric) Kane-Mele Hamiltonian.

In fact, this observation is also supported by experimental data when studying the $(0, 0) - (1e, 1h)$ charge transition in a similar gate configuration. Figure R4b below shows a close-

up of the $(0, 0) - (1e, 1h)$ triple point where two transitions (labeled α and β), separated by $\Delta E \approx 140 \mu\text{eV}$ can be observed. The transition scheme for α and β is depicted in Fig. R4d: For the creation of one electron and one hole via the interdot transition, a valence band electron of the first QD is excited into a (spin and valley matching) conduction band state of the second QD. The energy difference between the possible transitions of α and β is precisely the sum of the SO gap in the valence and conduction band (see transition schematic). The extracted energy of $\Delta E = 140 \mu\text{eV}$ matches well with $2\Delta_{\text{SO}}$ ($\Delta_{\text{SO}} \approx 70 \mu\text{eV}$) reported in the main text, suggesting that electron-hole symmetry is entirely preserved in this system. Please note, that Fig. R4 is part of a larger manuscript on electron/hole symmetry and spin-valley Pauli blockade physics for the $(0, 0) - (1e, 1h)$ charge transition, which is currently under preparation. Hence, we do not add the figure to the supporting material. Thank you for your understanding.

Figure R4: **a** Overview charge stability diagram showing the addition lines of the electrons to the left and right QD (yellow and purple lines), as well as the Coulomb resonances corresponding to a hole QD forming between the two electron QDs (see red lines). **b** Zoom-in for the first electron-hole triple point measured at $V_{\text{SD}} = 1 \text{ mV}$ (white circle in a). Two resonances, labeled α and β are observed. **c** detuning cut through the triple point shown in b. Two resonances separated by $140 \mu\text{eV}$ are observed. **d** Single particle spectrum of the first electron and the last hole orbital. For the $(0,0)-(1e,1h)$ interdot transition, an electron is elevated from the hole shell into the electron shell. When assuming spin and valley conservation, two possible sets of transitions, separated by $2\Delta_{\text{SO}}$ (α and β) are observed.

Reviewer 2:

The manuscript contains an experimental study of the low energy spectra in double bilayer graphene quantum dots. The low temperatures (10mK) allow to resolve very low energies. The valley-Zeeman, resulting from the topological orbital magnetic moment, result in large g-factors, which seem promising in the field of valleytronics. The paper is very well written, the graphs clearly illustrate the effects they measure and the experimental setups are quite well designed, providing good quality results.

We thank the reviewer for taking the time and effort to read and review our manuscript. We are delighted to read that the reviewer has a very positive reception of our experiment, the data quality, as well as the manuscript.

However, I do still have a few concerns:

There are other sources of zero-field splitting not included in the model, originated from symmetry breaking potentials, as Rashba or principal-plane asymmetry spin-orbit coupling. However, the experimental setup suggests that these should not be negligible, owing to different symmetry-breaking structures, gates, electric fields, etc.

Could the measured DeltaSO correspond to some combination of intrinsic and structural SO coupling? How does Deltakk' relates to structural SO coupling?

It would help to understand why DeltaSO differs from the one measured in other experiments. 'Previous experiments' have consistently (since 2012) shown that the intrinsic DeltaSO of about 45 micro-eV, which has been reproduced in several other samples using electron-spin resonance. The inter-valley term is, however, disorder induced or sample dependent, for which an upper bound is given. That would make this manuscript fall a bit short on novelty, if that was the main focus of the paper.

We thank the reviewer for raising this very important concern. Before going into detail on sources of extrinsic Bychkov-Rashba and principal-plane asymmetry (or pseudospin inversion asymmetry) SO coupling arising from inversion symmetry breaking or other extrinsic sources, we would like to comment on the novelty of our work. Our experiment is the very first to report on the magnitude of both Δ_{SO} and $\Delta_{KK'}$ of a confined single-electron in bilayer graphene, showing for the first time all details of the single-particle spectra of graphene QD. We consider our results highly relevant for the implementation of spin and valley qubits in such systems.

We would like to stress that to the best of our knowledge there is only one previous experiment, which have probed the SO gap in graphene in 2019 (J. Sichau et al. PRL **122**, 046403 (2019) and one experiment, which provided information on the SO gap in BLG in 2020 (L. Banszerus et al. PRL **124**, 1777701 (2020)). In contrast to this earlier work we show in this work for the first time the presence of the SO gap in the spectra of a single-electron QD, providing direct insights to the spin-valley texture of the low energy spectra. This not only allows to probe the SO coupling at very low k -values but also provides a quite high energy resolution for directly extracting this overall small energy gap. While the reviewer is correct

in stating that $\Delta_{KK'}$ is subject to external disorder, we now *consistently* show that $\Delta_{KK'}$ is smaller than our energy resolution in two different devices (see new Supplemental Fig. S3) and much smaller than what has been observed in carbon nanotubes before, making BLG QDs an interesting platform for valley based quantum information processing.

We fully agree with the referee that in our originally submitted manuscript the question on why the system can be well-described by the Kane-Mele type coupling term alone has not sufficiently been discussed in the text. In fact, in order to open up a band gap in BLG, the inversion/sublattice symmetry is explicitly broken by an external electric field, which potentially can give rise to Bychkov-Rashba SO coupling and principal-plane asymmetry (or pseudospin inversion asymmetry) SO coupling. The breaking of inversion symmetry by electric fields has been considered already by Kane and Mele (C.L. Kane and E.J. Mele, PRL **95**, 226801 (2005)), who estimate the Bychkov-Rashba SO gap in single-layer graphene to be smaller than $1 \mu\text{eV}$, even at large electric fields. For bilayer graphene S. Konschuh et al. present a detailed study on different types of SO coupling (S. Konschuh et al. PRB **85**, 115423 (2012)). Konschuh et al. find that for broken inversion symmetry Rashba terms remain very small and act only on the high energy bands of BLG. Close to the K-points of the BLG low energy bands, the Rashba terms vanish entirely and only Kane-Mele SO coupling is present.

Concerning the pseudospin inversion asymmetry (PIA) SO coupling, we recently had an interesting discussions with Jaroslav Fabian. It is indeed present when placing graphene on hexagonal boron nitride (K. Zollner et al. PRB **99**, 125151 (2019)) and according to J. Fabian the very same mechanism holds for BLG. However, this SO coupling is rather weak in fully encapsulated samples (see. e.g. Table III in K. Zollner et al. PRB **99**, 125151 (2019)) and even more important (according to eq. (6) in K. Zollner et al. PRB **99**, 125151 (2019)) this term vanishes completely for $k = 0$, being our point-of-operation for single-electron QDs. Overall this brings us to the conclusion that PIA can be also neglected in our case.

Please note that for quasi-zero-dimensional electronic systems, such as QDs, the wave-function is distributed very closely around the K and K' -point, which is why Rashba SO coupling is strongly suppressed. In the revised manuscript, we make this now more clear. Moreover, we include now also experimental data to verify this theoretical considerations. In particular we show now data of the experimentally extracted SO gap for a total of three devices at a number of different applied electric displacement fields (see new Fig. 5 in the manuscript, i.e. Fig. R5 below). Note that by changing the electric displacement field, we modify the strength of the inversion symmetry breaking, the size of the band gap, as well as slightly the shape of the confinement potential. Nevertheless, the SO gap observed in the device stays constant in the margin of errors with a mean value of $\Delta_{\text{SO}} \approx 60 \mu\text{eV}$.

Indeed the observed SO gap in our experiment, as well as in a recent study on one-dimensional confinement BLG quantum point contacts (L. Banszerus et al. PRL **124**, 1777701 (2020)), $\Delta_{\text{SO}} \approx 40 - 80 \mu\text{eV}$ are both larger than the values predicted for 'intrinsic' BLG by theory, as well as the values observed e.g. in 'bulk' single layer graphene probed by electron spin resonance measurements (J. Sichau et al. PRL **122**, 046403 (2019)).

The displacement field dependent measurements of Δ_{SO} , as well as the observation that electron-hole symmetry remains perfectly preserved (see last point by Reviewer 1 and Fig. R4 on page 11) both demonstrate that the SO coupling can be described by a Kane-Mele type

coupling term. The strength of the SO coupling term seems to be slightly increased which could have a number of possible origins: Firstly, symmetric proximity coupling between BLG and the top and bottom hBN may lead to an enhancement of the Kane-Mele term (L. Banszerus et al. PRL **124** 1777701 (2020)). Secondly, the influence of phonon-mediated enhancement of the SO coupling in BLG (H. Ochoa et al. PRB **86** 245411 (2012)) has not been considered by S. Konschuh et al., which could play a role. Finally, it is also worth mentioning that J. Sichau et al. have measured the SO gap in unconfined geometries, where the wavefunction is distributed over an extended region in k -space. The region close to the K and K' -point, where our experiment takes place has a couple of peculiarities: Firstly, as discussed above the Rashba terms vanish entirely close to the K and K' -point and secondly the Berry curvature is maximized around the K-points. The Berry curvature gives rise to a valley-dependent finite magnetization, which is responsible for the 'topological valley-magnetic moment' giving rise to the valley Zeeman effect. The valley magnetic moment will couple to the spin magnetic moment and either raise or lower a state in energy depending on whether the spin and valley magnetic moments are parallel or anti-parallel adding up to the 'intrinsic' tight-binding Kane-Mele term. The strength of this magnetic dipole-dipole coupling has not been studied experimentally or theoretically so far to the best of our knowledge, thus the magnitude is currently still unknown. We agree with the reviewer that this point has not been made sufficiently clear in the manuscript and should be included as inversion symmetry is explicitly broken. We have added a corresponding paragraph to the discussion section in manuscript which now reads (see page 7):

'Apart from the Kane-Mele SO coupling, which is intrinsically present in graphene and BLG, extrinsic Bychkov-Rashba SO coupling and pseudospin inversion asymmetry (or principal plane asymmetry) SO coupling can in principle also play a role [12-16,31,32]. The latter, which arises for example when placing graphene or BLG on substrates, such as e.g. hBN, depends on the magnitude of k (measured from the corners of the Brillouin zone) and is thus suppressed at the K and K' -points [16], where our devices are operated. The Rashba-type SO coupling needs to be discussed in more detail, since in our devices the inversion symmetry is explicitly broken by the applied out-of-plane displacement field. The breaking of inversion symmetry and the magnitude of the resulting Bychkov-Rashba SO gaps have theoretically been investigated by Kane and Mele for single-layer graphene [13] and by Konschuh et al. for BLG [12]. Both studies conclude that the Rashba type SO coupling is negligible ($\approx 1 \mu\text{eV}$) compared to the Kane-Mele coupling term. In addition, the Bychkov-Rashba SO coupling term is expected to be strongly suppressed in BLG single-electron QDs, since specifically in BLG this term vanishes for the low energy bands close to the K and K' -points [12]. Thus, all this is expected to lead to a displacement-field-independent SO gap that can be experimentally verified. In Fig. 5 we show the SO gap as function of displacement field, D . Here, Δ_{SO} has been extracted from zero B -field data similar to the measurements shown in Figs. 3a,d but with different back and split gate voltages such that the D -field is tuned from $D = 0.24 \text{ V/nm}$ to 0.34 V/nm , also resulting in different band gaps in the BLG as highlighted by the insets in Fig. 5. From all data presented in Fig. 5 – including data from a second single-electron DQD device (red triangles) and data from a different single-electron QD device (yellow square, more details in the supplementary information) – we conclude that the observed SO gaps are all consistent and within the error bars constant over the investigated D -field range, with a mean value around $\Delta_{\text{SO}} \approx 60 \mu\text{eV}$. From the absence of any dependency of Δ_{SO} as function of the strength of the potential breaking the inversion sym-

metry, we conclude that the experimentally extracted Δ_{SO} is dominated by the Kane-Mele coupling term. Interestingly, our Δ_{SO} values are slightly larger than what has been extracted in previous experiments performed in bulk graphene on trenched SiO_2 [18]. Also this value is larger than theory predicted [12], but might be explained by an enhancement due to phonon-assisted SO coupling [33]. In our case we expect that the SO coupling is slightly enhanced due to the proximity effect when encapsulating BLG with hBN crystals [17], very similar to the proximity enhanced SO coupling when placing BLG on WSe_2 [34].

Figure R5: Extracted value for Δ_{SO} as function of the applied electric displacement field. The red datapoints stem from the second DQD measured (see Fig. R1) and the yellow data point from finite bias spectroscopy measurements in a single quantum dot (see Fig. R2). This figure has been now included as new Figure 5.

Minor points:

* ϵ and ε are used for the same parameter, it seems.

* orthogonal axes epsilon and delta are introduced but delta is not used in the figures or Ms, rendering it unnecessary.

We thank the reviewer for both observations. We have corrected the manuscript and now only use ε throughout the text. We kindly disagree with the reviewer that the δ -axis is not discussed and thus unnecessary. On the second half of page 5 the dependence of the transitions on the δ -axis is explicitly mentioned to identify transition (iv) as one originating from from an excited state of the first, i.e. left QD.

For making this more clear we also added on page 4 the following sentence:

This is supported by the absence of any δ dependence of the transitions.

Comment:

I wonder if the valley-Zeeman effect could have a connection to the relationship of orbital angular momentum and sub-lattice spin in graphene? Or could an intuitive explanation be made, why the valley g-factor is so large?

We would like to thank the reviewer for this interesting comment. The origin of the valley magnetic moment in bilayer graphene is indeed quite interesting and it is very specific to gapped (bilayer) graphene. Our understanding is that this out-of-plane 'topological' orbital magnetic moment is directly connected to the finite Berry curvature close to the K and K' -points (see Fig.1 in our manuscript and A. Knothe and V. Fal'ko PRB **98**, 155435 (2018)). More handwavy, one can argue that the finite Berry curvature leads to a self-rotation of the wave packet, which then results in a magnetic moment. To what extent this is connected to a "true angular momentum" related to sublattice imbalance in graphene, as discussed e.g. in M. Prada A, is not yet quite clear to us and would need some more discussion with our friends and colleagues working in the field of theoretical physics. Interestingly in the work of M. Prada the term "Berry curvature" does not appear. For more details on the connection between Berry curvature and the magnetic moment, including the size of the valley g-factor we refer to A. Knothe and V. Fal'ko PRB **98**, 155435 (2018).

It is also worth mentioning that the valley magnetic moment in BLG has an entirely different origin than the valley magnetic moment in carbon nanotubes (CNTs). In CNTs the magnetic moment associated with the valley degree of freedom is actually not an independent degree of freedom but coupled to the orbital motion of the wavefunction around the CNT (F. Kuemmeth et al. Nature **452**, 448-452 (2008) and E. Laird et al. Rev. Mod. Phys. **87**, 703 (2015)). In CNTs, this orbital-like valley magnetic moment breaks the electron-hole symmetry at finite magnetic field (see Fig. 18 in E. Laird et al. Rev. Mod. Phys. **87** 703 (2015)), while perfect electron-hole symmetry is preserved in BLG (see Fig. R4).

Note: For electron-hole symmetry to be preserved the spin and valley have to be flipped when going from the electron to the hole side. This is true for BLG, while for CNTs, the spin is flipped but the valley remains the same.

REVIEWERS' COMMENTS

Reviewer #1 (Remarks to the Author):

I would like to thank the authors for their detailed response and manuscript revisions, which thoroughly address my questions and concerns. I think that the current version of the manuscript is suitable for publication in Nature Communications.

Reviewer #2 (Remarks to the Author):

The authors have made a great effort addressing all the concerns and comments risen by the referees in their reply. The last version with the new amendments contains, in my opinion, sufficient novelty to grant publication in Nature Comms. I thus recommend their manuscript for publication in Nature Communications.